

**Alternative Strategy for Estimating Zenith Tropospheric Delay**
**from Precise Point Positioning**
Jareer Mohammed * [a,b], Terry Moore[a], Chris Hill[a], Richard M. Bingley[c]
[a]Nottingham Geospatial Institute, University of Nottingham, UK
[b]Civil Engineering Department, College of Engineering, University of Wasit, Iraq
[c]NERC British Isles continuous GNSS Facility (BIGF), University of Nottingham, UK
**\*Corresponding author: jareermohammed@uowasit.edu.iq**
**Abstract:** This study considered zenith total delay (ZTD) estimation from precise point positioning
(PPP) based on GPS only (PPP GPS), GLONASS only (PPP GLO), and GPS+GLONASS (PPP
GPS+GLO) using both a conventional strategy when applying a model for the hydrostatic component
with an estimation of the wet component and an alternative strategy. The proposed alternative strategy
is to estimate both the hydrostatic and the wet components of the tropospheric delay using different
process noises with different mapping functions for both components in an extended Kalman filter
(EKF).  It was found that the receiver clock offsets and the estimated ambiguities would absorb some
errors in the ZTD when using the conventional strategy. The RMS values of the differences between
the double differenced (DD) GPS ZTD and the PPP ZTD, using the alternative strategy, were 6.5,
7.3, and 6.7 mm for PPP GPS, PPP GLO, and PPP GPS+GLO, respectively.  The results were
validated over one continuous week and then over one year. Validation was also performed through
comparison with the IGS ZTD values, for 12 weeks, with an overall RMS of 5.9 mm and against IGS
real-time products with an overall RMS of 8.1 mm. Furthermore, the alternative strategy also
provided significant improvements in the 5 cm convergence time in the vertical coordinate component
of the float ambiguity solutions to be on average, 51, 36 and 27 minutes for PPP GPS, PPP GLO and
PPP GPS+GLO solutions respectively.

**Keywords: precise point positioning (PPP), zenith total delay (ZTD), tropospheric models, real-**
**time ZTD**





## 1. Introduction

One of the major  sources of error affecting precise point positioning (PPP) (Zumberge et al., 1997) is the tropospheric delay of the GNSS signals. This is the delay of the signals as they propagate between the satellites and the user receiver, which is caused by the increased density of the troposphere. The delay is typically divided into wet and hydrostatic components. The total tropospheric delay is generally referred to as the zenith total delay (ZTD), which is derived from the individual slant delays using an appropriate mapping function. The zenith wet delay (ZWD) can be computed by simple subtraction of the zenith hydrostatic delay from the estimated ZTD. The hydrostatic delay is typically computed using atmospheric pressure data at the receiver, and the station's latitude and orthometric height (Saastamoinen, 1972). The atmospheric pressure can either be observed, obtained from numerical weather models or from climate datasets, such as the ERA-Interim products (Dee et al., 2011).

Many studies have been undertaken to improve the performance of ground-based GPS tropospheric delay estimation. Schueler et al. (2000) compared the ZTD estimated from their spatial interpolation of the tropospheric delay method with the IGS ZTD and found an agreement of 1.7 cm. Penna et al. (2001) compared the ZTD from the SBAS tropospheric model with that obtained from an analysis of one-year GPS carrier phase data analyzed and published by Dodson et al. (2000). They found that, for five stations in the United Kingdom, between 72% and 78% of the differences were <5 cm and between 96% and 99% were <10 cm. They also concluded that the RMS positioning errors in height component ranged from 4.0 to 4.7 cm, with maximum positioning errors ranging from 13.2 to 17.8 cm.

Leandro et al. (2006) presented and tested the UNB3m model as a modified version of the UNB3 model. They found that the predicted errors of the estimate of tropospheric delay from UNB3m had a mean value −0.5 cm and a standard deviation (STD) of 4.9 cm with respect to ray-tracing analysis. Furthermore, the treatment of the ZWD as a stochastic parameter, updated at every observation epoch in a Kalman filter, was found by Pany et al. (2007) to be a good tool with which to account for the high variability of the wet troposphere. Pace et al. (2010) presented a method for estimating ZTD residual fields using a ground-based GPS network. They modeled the zenith hydrostatic delay (ZHD) as an exponential function of latitude, whereas the ZWD was estimated every 5 minutes using a random walk stochastic model with a constraint of 20 mm/($\sqrt{h}$). They found that the ZTD residuals were of the order of 50–100 mm.



The results presented by Li et al. (2012), from 125 IGS stations during 2001-2005,
summarized the magnitude of the bias and RMS of the differences between IGS ZTD and the
estimated ZTD for the SBAS model (bias: 2.0 cm, RMS: 5.4 cm), UNB3 model (bias 2.0 cm, RMS:
5.4 cm ), UNB3m model (bias 0.7 cm, RMS: 5.0 cm), and IGGtrop model (bias: −0.8 cm, RMS: 4.0
cm). Their results showed that the new IGGtrop model provided the smallest bias and RMS errors.
However, the IGGtrop model is not available in the public domain and it still has a 4 cm RMS level
of uncertainty.
Numerical weather models that rely on meteorological data are used widely to estimate the
tropospheric delay. Yang et al. (2013) presented a new approach for estimating the slant tropospheric
delay from high-resolution numerical weather modeling products. Their RMS of the differences
between the tropospheric slant delay from the numerical weather model and the reverse computed
tropospheric slant delay from PPP was 6 cm below 10°, < 3.5 cm above 15° and 2 cm above 30°
elevation. Chen et al. (2014) mentioned that their model could predict the ZTD with average
uncertainties of about 2 cm under normal atmospheric conditions. Böhm et al. (2014) presented a new
blind tropospheric delay model (GPT2w) based on gridded values of water vapor pressure, a water
vapor reduction factor, and weighted mean temperature. They also compared their model with three
other blind models, referenced to the zenith total delay provided by IGS, as shown in Table 1.
**Table 1** Summary of the numerical weather model comparison by Böhm et al. (2014)

| Model | Mean bias of ZTD differences (cm) | Mean RMS of ZTD differences (cm) |
|---|---|---|
| SBAS | −2.50 | 4.55 |
| ESA (Krueger et al., 2005) | 0.83 | 3.82 |
| GPT2 (Lagler et al., 2013) | −0.28 | 3.79 |
| GPT2w | −0.02 | 3.61 |

All the tropospheric models used for providing the hydrostatic or wet components rely on
measured data to predict the tropospheric delay. However, they cannot account for weather variation
and thus, cannot provide highly accurate estimates of the tropospheric delay. Furthermore, none of
the tropospheric models account for the diurnal variations of the troposphere. For example, they
assume that pressure will be stable for a particular day of the year and that it will also be steady for



the same day from one year to the next. The assumption that the hydrostatic component will not
change during the day is likely to be flawed. However, the accuracy of all the ZTD results obtained
from the traditional models and the numerical models are within the range of centimeters, and those
within the range of <1 cm have a standard deviation of about 5 cm (UNB3m bias, 0.7 cm; RMS, 5.0
cm compared to the zenith total delay provided by IGS).
The objective of this study was to assess the accuracy of tropospheric delay estimates that are
achievable using different processing strategies. Most importantly, it considered a new alternative
strategy for the estimation of high accuracy tropospheric delay estimation, from PPP, in static and
real time situations.

## 2. Methodology

### 2.1. PPP Daily Solution Methodology

All the PPP solutions were processed using the POINT software, which was originally developed as
part of the iNsight project (www.insight-gnss.org). The POINT software is programmed in C++ and
its core is the EKF, as presented in Feng et al. (2008).
Undifferenced observations were used for each PPP daily solution using general observation
equations for the code and phase as follows:

For the pseudorange (m):

$$P_F^i = \rho^i + c\delta_{r\_code} - c\delta^i + \frac{I^i}{f_F^2} + \frac{S^i}{f_F^3} + T^i + M_{PF}^i + Q_{PF}^i + bias_{P,F} - bias_{P,F}^{\ i}$$

(1)

For the carrier phase (m):

$$L_F^i = \rho^i + c\delta_{r\_phase} - c\delta^i - \frac{I^i}{f_F^2} - \frac{S^i}{f_F^3} + T^i + m_F^i + q_F^i + \lambda_F(N_F^i)$$

(2)

where $i$ is the satellite index and $F$ represents the index of the GNSS frequency. For GPS satellites, $F$
= 1 (GPS $L_1$) and $F = 2$ (GPS $L_2$). For GLO satellites $F = 1$ (GLO $L_1$) and $F = 2$ (GLO $L_2$) with

$$f_{k\,L1} = f_{0L1} + k\Delta f_{L1}$$

(3)

$$f_{k\,L2} = f_{0L2} + k\Delta f_{L2}$$

(4)

Here, $k$ represents the frequency channel: $f_{0L1}$ = 1602 MHz for GLONASS $L_1$ band, $\Delta f_{L1}$ = 562.5
kHz frequency separation between the GLONASS carriers in the $L_1$ band, $f_{0L2}$ = 1246 MHz for
GLONASS $L_2$ band, and $\Delta f_{L2}$ = 437.5 kHz frequency separation between the GLONASS carriers in



the $L_2$ band. In the above, $\rho^i$ represents the geometric distance from the receiver to the satellite,
$c\delta_{r-code}$ is the receiver clock offset for code, $c\delta_{r-phase}$ is the receiver clock offset for phase, $c\delta^i$ is
the satellite clock offset,
$I^i$ is the first-order ionospheric bias term, $S^i$ is the second-order ionospheric bias term, $f_F$ is the GNSS
frequency, $T^i$ is the tropospheric bias. $M_F^i$ is the multipath error for pseudorange, $m_F^i$ is the multipath
error for carrier-phase, $Q_F^i$ is the noise for the pseudorange, $q_F^i$ is the noise for the carrier-phase.
$bias_{P,F}$ is the receiver code bias for pseudorange, $bias_{P,F}^i$ is the satellite code bias for pseudorange,
$\lambda_F$ is the wavelength, $N_F^i$ is the carrier phase ambiguity term.

9       For all the PPP daily solutions, a decoupled receiver clock (separate clocks for code and

carrier) is applied for both GPS and GLO (Collins et al., 2010), and the ionosphere-free observable
is used without applying any second-order ionospheric bias corrections. The ionospheric-free
combinations for the code and phase observables follow the process described by Dach et al. (2007).
The processing settings for the PPP solutions are summarized in Table 2.

15                    **Table 2** The processing parameters for PPP solution.

| Products (precise satellite coordinates and satellite clock offsets) | Natural Resources Canada (NRCan), unless otherwise mentioned. |
|---|---|
| Antenna phase centre offsets and variations | ANTEX from IGS, (Kouba, 2009). |
| Solid Earth tides, ocean tidal loading | Applied |
| Satellite Phase wind-up | Wu et al. (1993). |
| Pole and nutation motions | IERS conventions 2004 (McCarthy and Petit, 2004) |
| Carrier phase ambiguities | Float solution. |
| Cycle slip detection | Liu (2011) |
| Troposphere | Using Saastamoinen model for the hydrostatic component and estimate the wet as a state, unless otherwise mentioned. |
| Troposphere mapping function | New Mapping function (Niell, 1996) because of its capability for providing separate wet and dry mapping functions. |
| Tropospheric gradient | Chen model (Meindl et al., 2004), and using the Chen mapping function (Chen and Herring, 1997) . |
| Differential Code Bias | CODE (Dach et al., 2007). |



| Weighting function | No weighting functions are applied to the observations, except for the observations noise that is needed for the EKF, which is set to 2.0 m for the pseudorange measurements and to 0.01 m for the carrier phase measurements for both GPS and GLO. |
|---|---|

## 2.2. Global DD GPS Daily Solutions Methodology

For the UK stations, the processing strategy for the global DD GPS daily solutions is summarized in Table 3. Approximately 150 continuous GNSS stations (CGNSS) in the British Isles, including 100+ that are part of the Ordnance Survey of Great Britain (OSGB) national network, were included in the processing along with some 200+ IGS stations.

**Table 3** The processing parameters for double difference solutions.

| Software | Bernese GNSS Software version 5.2 (Dach et al., 2007) (Dach et al., 2009). Based on LSQ approach |
|---|---|
| Products (precise satellite coordinates and satellite clock offsets) | C13 (CODE repro2/repro_2013) re-analyzed satellite orbit and earth orientation parameter products |
| Satellite and receiver antenna phase center offsets and variations | I08.ATX models for antenna phase center variations |
| Troposphere | a-priori modeling of troposphere effects using VMF1G and estimation using zenith path delay and gradient parameters. |
| Ionosphere | mitigation of the first- and higher-order (second- and third-order and ray bending) ionospheric effects |
| Solid earth tides, Ocean tidal loading, and Atmospheric tidal loading | Applied |
| Carrier phase ambiguities | Fixed ambiguity. |

## 2.3. New Strategy for Estimating Tropospheric Delay

The new PPP processing strategy is to estimate both components of the troposphere (hydrostatic and wet) without relying solely on a tropospheric model for the hydrostatic component. The partial





derivative used in the design matrix is derived from the mapping function for the dry and wet
components, respectively:

$$\frac{\partial d_{trop}}{\partial d_h} = m(\varepsilon)_h \qquad (5)$$

$$\frac{\partial d_{trop}}{\partial d_w} = m(\varepsilon)_w \qquad (6)$$

where $m(\varepsilon)_h$ and $m(\varepsilon)_w$ are the mapping functions for the hydrostatic component and the wet
component respectively.

5        To allow the estimation process to properly reflect the characteristics of the two components of

the troposphere, the process noise for the two components has to be chosen carefully. The hydrostatic
component of the tropospheric delay is relatively stable and predictable, whereas the wet component
is more variable and less predictable since it depends on the distribution of water vapour in the
atmosphere. A higher process noise for the wet component allows the value to converge more quickly
even if the initial value is not accurate, and it allows the value to change rapidly in response to real
physical changes.

12       Since the wet and dry mapping functions and hence the partials used to estimate the components

are similar, it is not expected that the estimation process will determine reliable values for these two
components, but we expect that the sum of these two estimated components will reflect the ZTD
accurately. Therefore, the process noise and the initial values had to be chosen quite carefully in order
to provide a balanced, and optimal performance from the processing.

17       This tropospheric estimation strategy does not rely on the availability of meteorological data

or models. The values for the random-walk process noise were chosen empirically as 1e-5 m/$\sqrt{sec}$
and 3e-5 m/$\sqrt{sec}$ for the hydrostatic and wet components, respectively. We treat the hydrostatic
components as datum for the ZTD and the wet component will be the variation component of the
ZTD. Thus, we will end up with a ZTD with a good consistency with the ZTD from the independent
DD processing. Bear in mind that the validation will only be addressed by the ZTD components
because at this stage we are not attempting to separately validate the two components.
**3.  Datasets for Comparing the Two Methods**
For the purposes of this study, a data set focusing on the 100+ OSGB CGNSS stations that have daily
RINEX observation data files archived as part of NERC British Isles continuous GNSS Facility



(BIGF) and that were included in the global DD GPS daily solutions created by BIGF was chosen.
The locations of the CGNSS stations are illustrated in Figure 1.

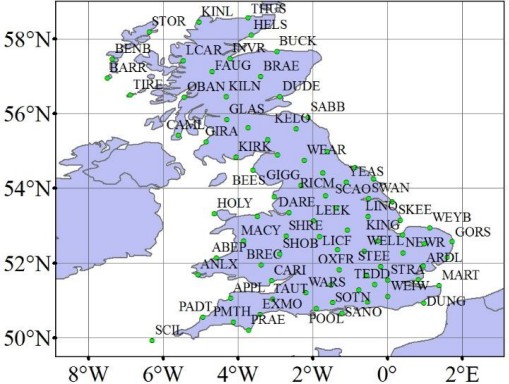

**Fig. 1** The 100+ OSGB CGNSS stations included in BIGF that were used for the assessment of the
new tropospheric strategy in this study
A 7-week period (detailed in Table 4) was used for the evaluation of the tropospheric delay using the
different tropospheric models. Only those OSGB CGNSS stations that operated continuously for a
specific GPS week with optimum 24-h observations recording each day and that were also present in
the global DD daily solutions were included. This led to the availability of 56–85 OSGB CGNSS
stations per week of the analysis, as detailed in Table 4.
**Table 4** GPS week and the number of OSGB CGNSS stations considered in the analysis for each
week.

| Calendar days | GPS Week | No. of CGNSS Stations |
|---|---|---|
| 12-18/1/2014 | 1775 | 56 |
| 19-25/1/2014 | 1776 | 85 |
| 26-01/2/2014 | 1777 | 79 |
| 02-09/2/2014 | 1778 | 74 |
| 09-15/3/2014 | 1783 | 74 |
| 16-22/3/2014 | 1784 | 76 |
| 23-29/3/2014 | 1785 | 79 |



**3.1. Results using the Traditional Strategy**
The 'traditional' PPP processing strategy uses a model for the hydrostatic component of the
tropospheric delay and estimates a correction to an initial hydrostatic component, along with the other
unknowns of the state vector. In this study, we have compared the effect of different models for the
hydrostatic component, and to compute the initial wet component. The models used included the
Saastamoinen model (Saastamoinen, 1973), UNB3m, which is a modified version of the University
of New Brunswick's (UNB3) neutral atmosphere model (Leandro et al., 2006), GPT as presented by
Boehm et al. (2007) and the SBAS model, which is one of the most commonly used tropospheric
delay models (RTCA, 1996). The processing was performed as a static solution for every 24 h with
an elevation cut-off angle of 10°.

11       To analyze the accuracy of the PPP solution, the RMS of the daily PPP difference from the

DD GPS for all coordinate components were computed for the 7-week data set and for all situations
using PPP GPS, PPP GLO, and PPP GPS+GLO. For the assessment of the accuracy of the
determination of the tropospheric delay, the daily mean of the differences between the common
epochs from the $ZTD_{DD}$, which is estimated every 1 hour, and $ZTD_{PPP}$ ($ZTD_{PPP} - ZTD_{DD}$) was
computed with the overall mean and overall RMS, as detailed in Table 5.
**Table 5** RMS of the daily differences of coordinates and the differences of the ZTD

| Model | Daily coordinate differences (RMS) (mm) | | | Tropospheric ZTD differences (mm) | | |
|---|---|---|---|---|---|---|
| | E | N | U | Mean | STD | RMS |
| GPS | | | | | | |
| UNB3m | 3.7 | 3.1 | 7.5 | 45.8 | 9.9 | 46.8 |
| GPT | 3.7 | 3.2 | 7.1 | 75.7 | 7.7 | 76.1 |
| Saastamoinen | 3.6 | 3.1 | 7.6 | 58.3 | 17.3 | 60.8 |
| SBAS | 3.6 | 3.1 | 7.5 | 58.4 | 16.0 | 60.6 |
| GLO | | | | | | |
| UNB3m | 3.7 | 4.9 | 11.5 | 44.5 | 11.1 | 45.9 |
| GPT | 3.7 | 4.9 | 12.7 | 75.0 | 8.7 | 75.5 |
| Saastamoinen | 3.7 | 4.9 | 10.7 | 57.0 | 18.9 | 60.1 |
| SBAS | 3.7 | 4.9 | 10.8 | 57.2 | 17.8 | 59.8 |
| GPS+GLO | | | | | | |
| UNB3m | 3.6 | 3.9 | 8.3 | 44.9 | 10.5 | 46.1 |
| GPT | 3.7 | 3.9 | 8.6 | 74.5 | 8.0 | 74.9 |
| Saastamoinen | 3.8 | 4.0 | 7.8 | 57.5 | 17.8 | 60.1 |
| SBAS | 3.7 | 3.9 | 7.9 | 57.1 | 16.2 | 59.4 |





Table 5 shows the difference in the tropospheric models used for comparison with the ZTD from DD

GPS. These results are consistent with the literature, e.g. Li et al. (2012) quote RMS differences of

5.4 cm, 5 cm and 4 cm for SBAS, UNB3m and IGGtrop respectively and Pace et al. (2010) quote

residuals in the order 50-100 mm.

### 3.2. Results Using The New Strategy:

To test the reliability of the alternative strategy for tropospheric estimation, there is a need to conduct

reasonable validations. All the following validations were conducted using OSGB or IGS stations in

the GDD GPS solutions as a reference value for the comparison. In theory, the tropospheric ZTD

from the GDD GPS represents a 'truth' value for the tropospheric delay. The reason is that it was

computed using a network of stations capable of estimating tropospheric ZHD and ZWD between

stations.

The validations were conducted using 7 consecutive days and regional validations at 56 OSGB

stations using the PPP GPS, PPP GLO and the PPP GPS+GLO. Further, a long-term validation was

also conducted using 22 OSGB stations for one year and using IGS stations.

### 4. Validation of the New strategy

### 4.1. Validation for One Continuous Week

To validate the new troposphelric strategy, it is important to test it over consecutive days to evaluate

its ability to provide an accurate tropospheric ZTD. It is also interesting to use zero values for

initializing the tropospheric ZHD and ZWD in PPP to test if the alternative tropospheric strategy with

the used mapping functions is capable of providing an accurate tropospheric ZTD independent of the

initial values. Therefore, this strategy was tested using 56 OSGB stations for seven consecutive days

(one continuous solution not 7 x daily solutions) on DOY 12-18, 2014 (GPS week 1775) starting with

zero initial values for tropospheric ZHD and ZWD. Figure 2 illustrates a sample of the ZTD estimates

from one of the 56 stations for one continuous week. The degree of consistency between the estimated

ZTD which is the sum of the estimated hydrostatic and wet components using the new strategy and

the ZTD derived from DD GPS can be clearly seen and the results are summarized in Table 6. Bear

in mind that this is a special example for one continuous week starting from zero initial values for the

wet and the hydrostatic components using the new strategy.

31





1  **Table 6** RMS of the daily differences of coordinates and the differences of the ZTD for one

2  continuous week using 56 stations.

| Model | Daily difference (RMS) (mm) | | | Tropospheric difference (mm) | | |
|---|---|---|---|---|---|---|
| | E | N | U | Mean | STD | RMS |
| | GPS | | | | | |
| The new strategy | 1.9 | 1.6 | 14.1 | -6.3 | 5.8 | 8.5 |

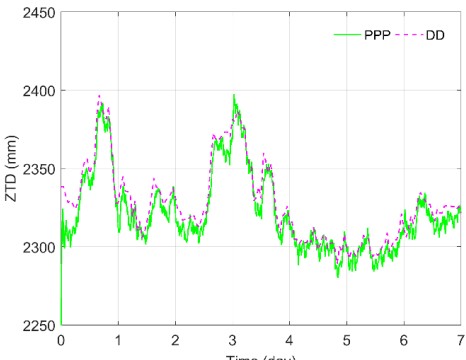

**Fig. 2** Comparison between ZTD from DD GPS and PPP GPS for the ASAP station for one
continuous week
It is evident from Figure 2 that the tropospheric ZTD using PPP GPS, which comes from the
estimated tropospheric ZHD and ZWD, agrees strongly with the tropospheric ZTD from the GDD
GPS. This alternative tropospheric strategy achieves this without using a model or any assumptions,
nor any meteorological data. The selection of the initial nominal zero values for the hydrostatic and
dry components was performed to establish whether the estimated values would converge to the
correct ZTD value which in this case was taken as the ZTD value obtained from the DD GPS. In
practice, as this will affect the convergence time, it is important to choose a suitable initial value, thus
for future processing, more realistic initial values will be used. This approach also confirmed the
suitability of the chosen mapping functions as well as the values selected for the process noise.



## 4.2. Regional Validation

To further evaluate the new processing strategy, it was important to process the same data set that had previously been processed when assessing the different tropospheric models (Table 4). Similar PPP configurations were adopted, except that the tropospheric delay was estimated using the new strategy with initial values of 2.1 m and 0.1 m for the dry and wet components, respectively. The PPP RMS of the daily difference is shown in Table 7 with a comparison of the tropospheric ZTD between the PPP situations and DD GPS.

**Table 7** RMS of the daily difference of PPP solutions from the DD solutions for 7 weeks for all stations

| Model | RMS Daily difference (mm) | | | Tropospheric difference (mm) | | |
|---|---|---|---|---|---|---|
| | E | N | U | Mean | STD | RMS |
| The new | GPS | | | | | |
| | 3.6 | 3.1 | 8.3 | -2.2 | 6.2 | 6.5 |
| strategy | GLO | | | | | |
| | 3.7 | 4.8 | 7.3 | -3.5 | 6.4 | 7.3 |
| | GPS+GLO | | | | | |
| | 3.9 | 3.9 | 7.1 | -3.9 | 5.4 | 6.7 |

Again, this table can be compared with Table 5 (both Tables 5 & 7 are referring to the same datasets of Table 4, the only difference between them is the tropospheric strategy). It is clear from Table 7 that the ZTD derived from the estimated hydrostatic and the estimated wet components compares well with the values obtained from DD GPS. Thus, the new strategy provides a value of the ZTD without the need for models except for the mapping function models for the partial derivatives, assumptions for the hydrostatic or wet components, or even the meteorological data. Moreover, the rate at which ZTD estimates can be computed is a direct function of the observation rate. In our case, the 30 second observational epochs meant that ZTD values were available every 30 second.

## 4.3. Long-Term Validation of the New Strategy

The tropospheric delay varies daily and seasonally. Therefore, it is important to validate the new processing strategy for deriving the tropospheric delay over a longer term. To achieve this, data from day of year (DOY) 2 (02/01) to DOY 365 (31/12), of 2014, from six CGNSS stations was used for a complete one-year analysis, under the same PPP configuration, using the new strategy. For each





station, the mean and the standard deviation of the differences between the ZTD estimates from PPP
and DD were calculated. A summary of the results is shown in Figure 3.
**Fig. 3** One-year tropospheric mean differences (marker) from DD GPS and RMS differences (line)
from DD GPS for six stations and all PPP cases.

8         The accuracy of the tropospheric delay estimations achievable for these stations can be seen

in Figure 3. This illustrates that the new strategy may be used at any time of the year, irrespective of
the weather conditions. Most importantly, the suitability of PPP GLO for the tropospheric delay
estimation is clearly highlighted, since the performance of the PPP GLO situation appears no different
from the PPP GPS situation. In addition, this gives the potential for PPP users to create two solutions
for the same station, instead of one solution of PPP GPS or PPP GPS+GLO.
**4.4. Global Validation using IGS Stations**
For the validation of the new processing strategy over different locations, a number of IGS (Dow et
al., 2009) stations were chosen, as illustrated in Figure 4, based on two conditions. Firstly, they had
to be available for one continuous week to ensure continuity, and secondly, their solution had to be
available from the IGS for the station coordinate estimation and, most importantly, for the final
tropospheric solution. The chosen stations ranged between −68.57° and 74.7° latitude, −176.6° and
174.8° longitude, and −27.99 and 2228.3 m height. This degree of variation was selected to provide
an indication of the effectiveness of adopting the new strategy for ZTD estimations.



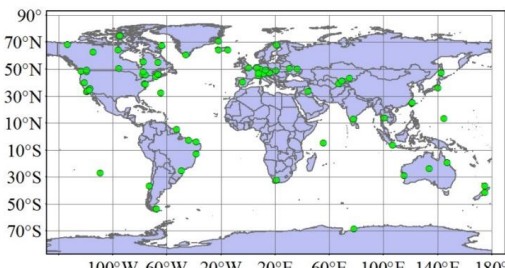

**Fig. 4** IGS stations used for the assessment of the new tropospheric strategy in this study
To ensure that the method was tested for different weather activity throughout the year, we choose
the first GPS week from each month. Table 8 shows the number of IGS stations available for each of
the selected GPS weeks in 2014.
**Table 8** GPS week and the number of IGS stations considered in the analysis of each week

| Calendar days | GPS week | No. IGS stations |
|---|---|---|
| 05-11/01/2014 | 1774 | 74 |
| 02-08/02/2014 | 1778 | 76 |
| 02-08/03/2014 | 1782 | 74 |
| 06-12/04/2014 | 1787 | 56 |
| 04-10/05/2014 | 1791 | 64 |
| 01-07/06/2014 | 1795 | 66 |
| 06-12/07/2014 | 1800 | 64 |
| 03-09/08/2014 | 1804 | 71 |
| 07-13/09/2014 | 1809 | 73 |
| 05-11/10/2014 | 1813 | 74 |
| 02-08/11/2014 | 1817 | 68 |
| 07-13/12/2014 | 1822 | 70 |

PPP daily solutions were compared with the IGS weekly solutions to overcome the variations
of the IGS daily solutions. Two products (orbits, clocks) were used: the IGS final products and the
NRCan products. For the ZTD comparison, two products are available from IGS: one is for near real
time, which has 5-min intervals, and the other is the final product produced after 4 weeks, which is
at 2-hour intervals. The estimated ZTD values for the common epochs using the new strategy were
compared with both the IGS near-real-time (NRT) and the IGS final ZTD (Final) products and
summarised in Table 9.



**Table 9** RMS of the daily difference of PPP solutions from the IGS weekly solutions for 12 weeks
for all stations

| Tropospheric Delay Method | ZTD reference value | Daily difference (RMS) (mm) | | | | Tropospheric difference (mm) | | |
|---|---|---|---|---|---|---|---|---|
| | | Orbits & Clocks | E | N | U | Mean | STD | RMS |
| The New strategy | NRT | NRCan | 4.0 | 3.8 | 10.0 | 1.1 | 5.8 | 5.9 |
| | Final | NRCan | | | | 2.0 | 5.8 | 6.1 |
| | NRT | IGS_final | 7.9 | 4.5 | 18.7 | 1.1 | 5.8 | 5.7 |
| | Final | IGS_final | | | | 2.0 | 5.9 | 5.5 |

### 4.5. Global Validation using Real-Time Products

The new strategy was evaluated in a pseudo *real time* situation; the same IGS data sets as considered previously in Table 8 were used. The PPP processing strategy was the same as before, except it was performed using IGS real-time IGS01 satellite ephemerides and satellite clock offsets. The real-time products were not available for DOY 10, 11, 128–130, 154–158, 191–193, 279–284, 306–312, and 341–345, and therefore these days were excluded. The estimated ZTD from the PPP using the new strategy was compared with the final and near-real-time ZTDs.

**Table 10** RMS of the daily difference of PPP solutions from the IGS weekly solutions using real-time products

| Tropospheric Delay Method | ZTD reference Value | Orbits & Clocks | Daily difference (RMS) (mm) | | | Tropospheric difference (mm) | | |
|---|---|---|---|---|---|---|---|---|
| | | | E | N | U | Mean | STD | RMS |
| The New strategy | NRT | IGS real-time | 20.3 | 15.7 | 27.7 | 1.3 | 8.0 | 8.1 |
| | Final | | | | | 2.0 | 7.8 | 8.1 |



It is important to highlight that these results compare the final IGS ZTD values to those based
on real-time PPP. This validation clearly shows that the real-time results are almost as good as the
post-processed final results mentioned in Table 8, and therefore that the new strategy may be used
for PPP in real time when users have access to the real-time products, without the need to implement
a tropospheric model except for mapping functions.

## 7    5.  Results Discussion of The Two Methods

It is important to explain why the tropospheric models do not have the ability to produce accurate
ZTDs, but do have a consistent level of accuracy among them. Zhang and Gao (2001) mentioned that
the consistency of the position vector could reflect the quality of the troposphere delay estimates.
However, the results in Tables 5 and 6 show this is not necessarily true for the quality of the
tropospheric delay in the PPP case. At this level of accuracy, the position vector can be estimated
precisely while the estimated troposphere will not be estimated precisely because the receiver clock
offset and the estimated ambiguities absorb some of the error in the tropospheric delay estimation, so
the coordinates are not affected.
Considering the results in Table 5, when using a model for the hydrostatic component and
estimating the wet delay, the receiver clock will absorb some of the effect of uncertainty in the
modelled hydrostatic component. This is evident in Figure 5, which illustrates the differences between
the estimated carrier receiver clock from PPP GPS and the estimated carrier receiver clock from PPP
GPS using the new strategy for three sample stations in the UK: DRUM (Drumalbin), FARB
(Farnborough), and KEYW (Keyworth) for DOY 12 (12/01), 2014.

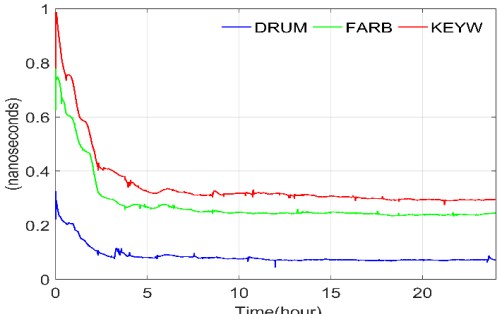

**Fig. 5** Differences between the estimated carrier receiver clock from the new strategy and the
carrier receiver clock from the traditional method



Figure 6 shows an example of phase residuals, the estimated ionosphere-free ambiguity and
the widelane ambiguity for satellite PRN 11, DRUM station using the tropospheric model for the
hydrostatic component with an estimation of the wet component (left) and using the new strategy
(right) for tropospheric estimation, respectively. It can be seen that there is neither an effect on the
phase residual nor the widelane ambiguity.

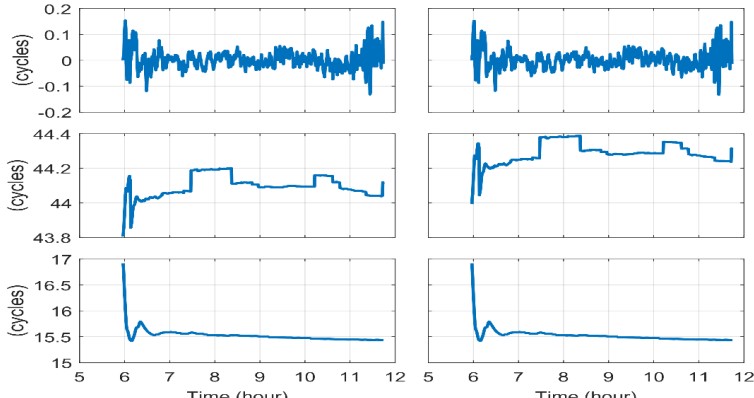

**Fig. 6** Tropospheric model effect on phase residuals (top), ionospheric-free-ambiguity (middle) and
the widelane ambiguity (bottom) using a model for the hydrostatic component with an estimation
for the wet component (left); using the new strategy for phase residual, ionospheric ambiguity and

12                    widelane ambiguity (right).

It can be concluded from this that the unmodeled tropospheric delay and the receiver clock,
as well as the ionosphere-free ambiguity, are linked to each other. In PPP applications that require
ambiguity fixing, an error in the ionosphere-free ambiguity, caused by unmodeled tropospheric delay,
will result in incorrect narrow lane ambiguities and hence an incorrect position solution. This is
because the narrow lane ambiguities are computed from the non-integer ionosphere-free ambiguities
and the integer-valued widelane ambiguities.





## 6. PPP Improvement When using the Alternative Tropospheric Strategy

The suitability of the alternative strategy for tropospheric ZTD estimation can also be reflected in the PPP performance in terms of the convergence time and repeatability as follows.

### 6.1. Convergence time

The efficiency of the new strategy for tropospheric estimation is also reflected in the PPP performance. It is known that the convergence time for the PPP solution is longer for the height component and this has a relation with the tropospheric estimation. Applying the new strategy not only gives high accuracy ZTD from the PPP solution but it also improves the convergence time. The convergence time was defined as the required time from the first epoch to reach to a chosen agreement with the '*truth values*' from the DD solution. A significant improvement can be seen in Figure 7, where the coordinate convergence time is shown for 20, 10, and 5 cm for the height component for all models and all PPP cases, using the dataset in Table 4. The improvement in the time taken to converge to 5 cm, compared to the current strategy, using UNB3m, GPT, Saastamoinen, and SBAS models were 49, 66, 44 and 43 minutes for PPP GPS (top), 31, 53, 30 and 29 minutes for PPP GLO (middle), and 28, 37, 22 and 23 minutes for PPP GPS+GLO (bottom), respectively.

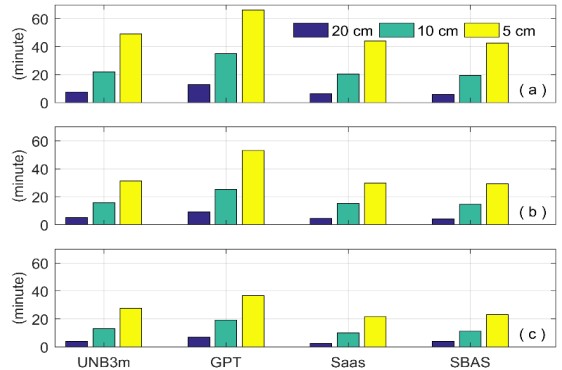

**Fig. 7** Improvements in the convergence time when using the new strategy for PPP GPS (top), PPP GLO (middle) and PPP GPS+GLO (bottom)



**6.2. The Improvement on one-year Repeatability**
Another potential improvement is in the repeatability over a long period. This can be seen in
Figure 8, which represents a comparison of two OSGB stations ALDB and AMER using the
conventional and alternative tropospheric strategies.

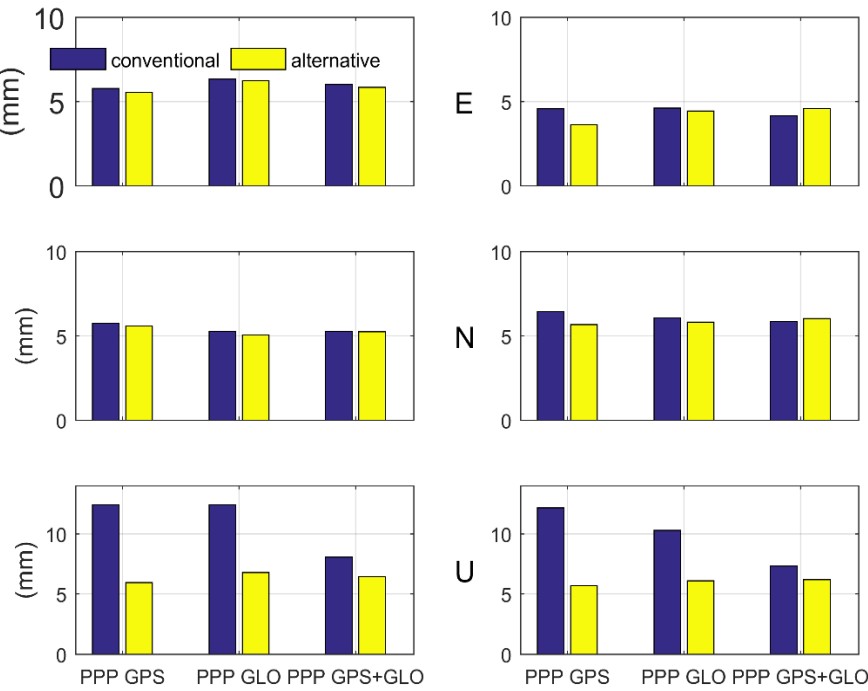

**Fig. 8** Comparison between the repeatability analysis when using a model for troposphere and
by using the alternative tropospheric strategy (AMER Left, ALDB Right)
From Figure 8, it can be seen that the main improvement obtained is in the repeatability of the station
coordinate Up component to be 51.9% (GPS), 45.4% (GLO) and 20.0% (GPS+GLO) for AMER
station, 53.2% (GPS), 40.5% (GLO) 15.4% (GPS+GLO) for ALDB station.
7.  **Conclusions**
This study set out to compare the effect of different models for the hydrostatic component and
to present an alternative method to improve the estimation of ZTD from PPP with post processing
and with *real- time* estimation, without using any external tropospheric information.





Based on the tropospheric analysis, it can be concluded that the traditional method for
estimating ZTD does not have the ability to produce millimeters accuracy of tropospheric delay, even
though the position estimation is accurate, because the receiver clock absorbs the unmodeled
tropospheric delay and the estimated ambiguity.
The presented method does not rely on any zenith tropospheric delay models for the
hydrostatic component. Instead, it separates the hydrostatic and the wet components using different
mapping functions and different process noise in an extended Kalman filter. Following this method,
the estimated $ZTD_{PPP}$ can be modelled more accurately and within millimeters accuracy compared to
the $ZTD_{DD}$. A series of validations were conducted for the alternative tropospheric strategy using PPP
GPS, PPP GLO and PPP GPS+GLO. Validation using 7 consecutive days were first performed,
followed by an expanded regional validation that was done for a seven week dataset of OSGB stations
in the UK, and then a long-term (over one year) validation for 22 OSGB stations. A global validation
using ~76 stations IGS stations was then done over a different period, and conducted in three stages,
using EMX final, IGS final and IGS real-time precise products. The estimated tropospheric ZTD
compared favorably with the IGS final and near real-time solutions. It was also proposed that this
approach can be used in *real-time* as well as in post-processing without a significant difference
between the results.
Lastly, it was shown that with the alternative tropospheric strategy, PPP users will not only
obtain an accurate tropospheric ZTD, but they will also have an improvement in the convergence
time and the repeatability of station height over the long term.
**Acknowledgment:** The first author acknowledges the financial support he received from his
government in Iraq during the period of his postgraduate research study at the University of
Nottingham.

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
