# Peer review of "Alternative Strategy for Estimating Zenith Tropospheric Delay 1 from Precise Point Positioning 2 3 Jareer Mohammed \* a,b, Terry Moorea, Chris Hilla, Richard M. Bingleyc 4 aNottingham Geospatial Institute, University of Nottingham, UK 5 bCivil Engineering Department, College of"

_Atmospheric Measurement Techniques, 2017_

## Referee Comment (RC1) · Anonymous Referee #1 · 8 Dec 2017

**General comments:**

The study describes an alternative strategy for estimating ZTDs from PPP. Whereas in the conventional approach the a prior zenith hydrostatic delay (ZHD) is fixed and the zenith wet delay (ZWD) is estimated (adjusted), in the alternative approach both the ZHD and the ZWD are estimated (adjusted). In fact, the conventional approach appears to be a special case of the alternative approach (if one puts a very tight constraint on the a prior ZHD in the alternative approach, i.e. fix the a prior ZHD, one ends up with the conventional approach). This point (see major comments) together with some other points (see specific comments) must be clarified.

Major comments:

(1) The conventional approach is a special case of the alternative approach (see my general comment). Is this true, or did i miss something? If this is true, please, write it down somewhere in the beginning of the manuscript.

(2) The reference solution is a DD solution (Table 3). This DD solution follows the conventional approach, i.e. the a prior ZHD (from VMF1G) is fixed and the ZWD is estimated. So, the reference for your alternative PPP approach is a conventional DD approach. Please write it down in section 2.2.

(3) Table 5 (the conventional approach) can be directly compared with table 7 (the alternative approach). Where do the huge biases and rmse ( $\sim$  6 cm) in table 5 come from? A blind model, e.g. the GPT2w, is much better (Table 1).

(4) In the alternative approach you also make use of an a prior ZHD. You also mention that the constraints for the ZHD and ZWD must be chosen 'very carefully'. I would like to see the following experiment: repeat the processing with various constraints for the ZHD and add the results in table 7. In particular, if you use some reasonable a prior ZHD (from GPT) and apply a very tight constraint on the ZHD you should obtain the results in table 5. In addition, i suggest to add the results for various constraints in Table 9 and 10.

Minor comments:

Page 3, line 13: In this context (slant tropospheric delays) you should add and comment on the following recent AMT paper:

Kačmařík, M., Douša, J., Dick, G., Zus, F., Brenot, H., Möller, G., Pottiaux, E., Kapłon, J., Hordyniec, P., Václavovic, P., and Morel, L.: Inter-technique validation of tropospheric slant total delays, Atmos. Meas. Tech., 10, 2183-2208, https://doi.org/10.5194/amt-10-2183-2017, 2017.

Page 3, line 21: '...All the tropospheric models used for providing the hydrostatic or
wet components rely on measured data to predict the tropospheric delay. However, they cannot account for weather variation and thus, cannot provide highly accurate estimates of the tropospheric delay. Furthermore, none of the tropospheric models account for the diurnal variations of the troposphere. For example, they assume that pressure will be stable for a particular day of the year and that...' Here you mean the blind tropospheric models mentioned in table 1? Please clarify this. Tropospheric models derived from weather models (VMF1, UNB-VMF1 etc.) or measurements take into account the diurnal variation, etc.

Page 5, Table 2:

Troposphere mapping function: Simply write 'New Mapping Function (Niell, 1996)'. There is no need to explicitly state that it provides separate wet and dry MF. This is standard.

Page 6, Table 3:

Troposphere: I suggest to replace 'a-prior modeling of troposphere effects using VMF1G and estimation using zenith path delay and gradient parameters.' by 'a-prior modeling of troposphere effects using VMF1G and estimating the zenith wet delay and gradient parameters.' This is the conventional approach. I also suggest to use the same items in Table 2 and 3 (one item for the tropsopheric mapping function and another item for the a-prior ZHD and ZWD).

Page 7, equation 5 and 6: Replace 'dtrop' by 'T' (see equation 1 and 2) and replace 'dh' by ZHD and 'dw' by ZWD.

Page 10, line 2: 'Table 5 shows the difference in the tropospheric models used for comparison with the ZTD from DD GPS. These results are consistent with the literature, e.g. Li et al. (2012) quote RMS differences of 5.4 cm, 5 cm and 4 cm for SBAS, UNB3m and IGGtrop respectively and Pace et al. (2010) quote residuals in the order 50-100 mm.' The results are inconsistent with many other studies. For example, the recent
study by Dousa et al 2016 AMT, shows that ZTDs from DD and PPP agree very well. I suggest to comment on that and add the following recent AMT reference:

Douša, J., Dick, G., Kačmařík, M., Brožková, R., Zus, F., Brenot, H., Stoycheva, A., Möller, G., and Kaplon, J.: Benchmark campaign and case study episode in central Europe for development and assessment of advanced GNSS tropospheric models and products, Atmos. Meas. Tech., 9, 2989-3008, https://doi.org/10.5194/amt-9-2989-2016, 2016.

Page 12, line 3: 'Similar PPP configurations were adopted, except that the tropospheric delay was estimated using the new strategy with initial values of 2.1 m and 0.1 m for the dry and wet components, respectively' Here, I strongly recommend to add additional experiments (see my major comment). For example, instead of 2.1 m and 0.1 m as initial values use the GPT, and instead of a very loose constraint for the ZHD use a very tight constraint for the ZHD, i.e. fix the ZHD. This procedure shows the transition from the alternative to the conventional approach.

---

## Referee Comment (RC2) · Anonymous Referee #3 · 11 Dec 2017

General Comments This paper intents to describe an alternative strategy to the conventional one used to estimate ZTD. The proposed alternative strategy is interesting and deserves to be evaluated. However, I think that the manuscript needs major revision before being ready for publication. The following aspects need to be improved. First, throughout the manuscript there is a lack of explanation of the obtained results. They are mainly presented in form of tables and figures with very short and poor text of critical discussion. Second, the proposed alternative strategy is supposed to improve both post–processed and Real-Time ZTD estimation. Post-Processed and Real-Time analysis have different requirements in term of latency and accuracy that should be discussed and considered. I have the feeling that the in the manuscript post-processed,

near-real time and real time issues, along with the related products used for data reduction and evaluation, are sometimes mixed up.

Below specific comments for each section of the manuscript.

Abstract Page 1 - line 17: Delete 'GPS' before ZTD Page 1 - line 18 Delete 'PPP' before GPS, GLO and GPS+GLO. How the differences are computed? Is it 'DD-PPP' or 'PPP-DD'? Page 1 - - line 19- 21: 'Validation was also performed through comparison with the IGS ZTD values, for 12 weeks, with an overall RMS of 5.9 mm and against IGS real-time products with an overall RMS of 8.1 mm' I think that in the first part of this sentence the authors are referring to 'IGS final ZTD estimates' (http://www.igs.org/products) while in the second part to the 'IGS real-time orbit and clock products' (http://www.igs.org/rts/products) delivered in the framework of the IGS Real-Time Service and used for Real-Time PPP. If it is so, the sentence has to be properly rewritten. Anyway a clarification is necessary.

Introduction Page 2 - line 15. '....an agreement of 1.7 cm' in terms of mean or standard deviation? Page 2 – line 27. Use the reference: B. Pace, R. Pacione, C. Sciarretta, G. Bianco, "Computation of Zenith Total Delay Correction Fields using Ground-Based GNSS estimates", IAG Symposia Series. vol 137-2012/ IAGS-D-13-00021 instead of Pace et al. (2010). Page 3 – Line 1-7. The authors should add a reference of the cited models along with a brief description of their characteristics and differences in order to let the reader to understand the results presented. Page 3 - Line 19. I suggest replacing Table 1 with a brief summary of the results of Böhm et al. (2014) that are relevant to the present manuscript. Page 4 - Line 8. '..static and real-time situation' I think it is post-processed and real-time situation. Page 4 - Line 10. Sentences on the organization and structure of the paper must be added.

PPP Daily Solution Methodology Page 5 - Line 15. Table 2. Please consider that: 1. ANTEX from IGS the proper reference to the ANTEX file is M. Rothacher, R. Schmid: ANTEX: The Antenna Exchange Format, Version 1.4, 15 September 2010,

ftp://igs.org/pub/station/general/antex14.txt Which antex file is used? It the same antex file used to process DD data of Table 3 in section 2.2? 2. Troposphere: Using Saastamoinen model for the hydrostatic component and estimate the wet as a state, unless otherwise mentioned. Please explain the meaning of 'state'. What is the ZTD sampling rate? 3. Troposphere mapping function: drop 'new'

Page 6 - Line 9. Table 3. Please consider that: 1. Troposphere. What is the ZTD sampling rate?

There are several differences in the models summarized in table 2 and table 3: mapping function, carrier phase ambiguities, and products. Please add a comment on these diversities and on their potential impact on the estimated ZTD.

New Strategy for Estimating Tropospheric Delay Page 7 - Line 18-19. On which ground the values for the random-walk process noise were chosen? Did the authors test other values?

Datasets for Comparing the Two Methods Page 8 - Line 10.'. . . with optimum 24-h observations' What do you mean with optimum?

Results using the Traditional Strategy Page 9 - Line 9. Is it 'static' or 'post-processed'? Page 9 - Line 17. I suggest to consider also the site coordinate repeatability as an internal quality metric to check the different solutions.

Validation for One Continuous Week Page 10 - Line 19. Typo 'troposphelric' Page 10 - Line 24. The authors are testing the new strategy over 1 week continuous solution, they are not averaging 7-daily solutions to get the weekly solution. A comment on how the GNSS orbits are handled is required since GNSS orbits over 1 week are available as 7 independent daily solutions with a possible jump at midnight between consecutive days. Page 11 - Line 1. Table 6. The ZTD agreement is -6.3 mm mean and 5.8 mm std. Is this a 'good' agreement' or there is room for improvement? In post-processing the expected agreement, in terms of std, between different sw/solutions is about 2 mm.

(see Pacione et al: Atmos. Meas. Tech., 10, 1689-1705, https://doi.org/10.5194/amt-10-1689-2017, 2017) Page 11 - Line 7. Fig. 2. Could the authors explain why PPP ZTD is systematically larger than DD ZTD? Are there any boundary problems in the PPP ZTD at the beginning and end of the 7-day period?

Long-Term Validation of the New Strategy Page 13 - Line 9 and Fig. 3. To demonstrate that there is no seasonal behavior in the proposed strategy, one average value for the whole year is not enough. One value per month or one value for season should be considered.

Global Validation using IGS Stations Page 14 - Line 11-13. Could you please better explain which products you used? According to what listed in http://www.igs.org/products IGS is not delivering NRT ZTD. Global Validation using Real-Time Products Page 16 - Line 5. I suggest the following references for Real time ZTD estimation and performance 1. Dousa J, Vaclavovic P (2014) Real-time zenith tropospheric delays in support of numerical weather prediction applications. Advances in Space Research (2014), Vol 53, No 9, pp 1347-1358, doi:10.1016/j.asr.2014.02.021 2. Ahmed F et al Comparative analysis of real-time precise point positioning zenith total delay estimates, 2014 GPS Solut. DOI 10.1007/s10291-014-0427-z

Conclusions Page 20 – Line 1-4. I think that what the authors are assessing is very strong. The new proposed strategy is interesting but needs to be checked and tested more deeply before drawing this conclusion.

References When possible, replace the reference listed as 'proceeding' with a peer review publication.

Other comments: 1. All the acronyms in the text have to be explained and the same abbreviation should be used to refer to the same thing, for example sometimes is used 'GLONASS' sometimes 'GLO'. 2. In the tables the sign of the considered difference has to be reported.

---

## Referee Comment (RC3) · Anonymous Referee #2 · 18 Dec 2017

In this manuscript Authors describe alternative strategy for estimating ZTD from precise point positioning method. The main assumption of proposed approach is to estimate both hydrostatic and wet part of ZTD using different process noise and different mapping functions for both components. Generally, this is very interesting research but some major remarks should be taken into account before this manuscript can be accepted for publication.

Major remarks and questions:

1. Authors presented results of ZTD estimation obtained using alternative strategy of PPP method and compered these results to ZTD obtained using double differenced

method. Why Authors did not compere their results also to the conventional PPP?

2. Line 24-26: It is not entirely true. Authors should remember about VMF, where different values of ZHD and ZWD are provided during the day.

3. In my opinion, in the introduction there is lack of information about VMF. Authors can also pay attention to VMF3/GPT3 (doi: 10.1007/s00190-017-1066-2).

4. Table 2: Which ANTEX version was used by the Authors? Why Authors did not fixed ambiguities?

5. In PPP solution Authors used precise products from NRCan while in DD products from CODE. Products from different sources can caused differences in parameters estimation, which may lead to wrong conclusions. Ephemerides and clocks should be used from the same sources, either from CODE or from NRCan. In my opinion it will be nice to see comparison between two PPP methods: first with conventional strategy, and second with alternative strategy proposed by the Authors. Of course in both solutions the same processing parameters and precise products should be used. Moreover, Authors should remember that for precise tropospheric parameters estimation the VMF1 is often used. Unfortunately Authors did not provide any information about it and did not used solution with VMF1 to comparison (e.g. in Table 5). I think that this is a big deficiency of presented manuscript. In the paper it is hard to find explanation for this, especially that VMF1 was used in DD solution.

6. One more question related to solutions using traditional strategy. In section 3 Authors presented used tropospheric models. Why Authors did not used GPT2 which is more precise than other presented models? Of course the VMF case should be also reconsider in this place.

7. I have also general comment to the all results presented in tables: 5, 6, 7, 9, and 10. I think that in case of ZTD differences it will be nice to see also maximum and minimum values. I also advise Authors to think about replacing mean values into a median.

[Figure]

8. In validation section it is not clear how the ZTDs from IGS and Authors strategy were compered. In IGS tropospheric products are available with 5 minutes interval. I think that information about estimation interval should be placed into manuscript (both for IGS and Authors solution), as well as information how the comparison of two products looked like. Was ZTDs comparison conducted at the same epochs or maybe some averaging was used? It should be clearly explained in the text.

9. Section 6.1: I have serious objections to the content of this section. Authors show how the 'new strategy' impacts on convergence time. However they only presented comparison to the low precise tropospheric models. I think that it is necessary to present results to more precise solution e.g. with VMF. Furthermore, Authors did not provide any information about processing strategies, or number of used stations. Readers may also have a problems with results interpretation. Are there mean value of convergence time in Figure 7? Or maybe these values are for one station? If there are mean values, Authors should present also RMS or STD parameters.

10. Section 6.2: In this section Authors compared only for two stations. Why exactly these? Authors present results for conventional and alternative approaches. Which tropospheric model was used in the conventional solution?

11. In presented by Authors solution the ZHD and ZWD components are estimating separately. In presented manuscript Authors presented only total value of ZTD. However this not mean that ZHD and ZWD are correctly estimated. For example there can be some biases for both component but with opposite signs. Thus in ZTD this error will not be visible. It should be notice that proper estimation of ZWD is crucial for many application, e.g. for conversion to IWV and analysis of atmospheric opacity is performed (doi: 10.1007/s10291-017-0675-9). Unfortunately in presented manuscript there it is not explained whether the ZWD can be directly used for such (or similar) application.

12. In presented manuscript there is also lack of ZHD and ZWD estimation errors analysis. I think that it is necessary to show how the values from covariance matrix

looks like during the processing time. Also it will be nice to see post-fit residuals. Of course, only examples for selected stations can be presented.

The proposed by Authors strategy is very interesting but it needs to be checked, tested, and verified more deeply before publication.

---

## Author Comment (AC1) · 3 Feb 2018

Please see attached a pdf file with our reactions on the comments in the review.

Please also note the supplement to this comment:
https://www.atmos-meas-tech-discuss.net/amt-2017-321/amt-2017-321-AC1-supplement.pdf
* * *

---

## Author Comment (AC3) · 3 Feb 2018

General Response:

We thank you for your time in reading the manuscript, and for the comments that have been followed to improve our manuscript. We have followed your advice as it will be shown below in the response to the comment and the mark-up version of the manuscript which have been also included.

Anonymous Referee #1
General comments:

The study describes an alternative strategy for estimating ZTDs from PPP. Whereas in the conventional approach the a prior zenith hydrostatic delay (ZHD) is fixed and the zenith wet delay (ZWD) is estimated (adjusted), in the alternative approach both the ZHD and the ZWD are estimated (adjusted). In fact, the conventional approach appears to be a special case of the alternative approach (if one puts a very tight constraint on the a prior ZHD in the alternative approach, i.e. fix the a prior ZHD, one ends up with the conventional approach). This point (see major comments) together with some other points (see specific comments) must be clarified.

Major comments:

(1) The conventional approach is a special case of the alternative approach (see my general comment). Is this true, or did i miss something? If this is true, please, write it down somewhere in the beginning of the manuscript.

Authors' response:
The conventional approach could be treated as a special case of the suggested approach because it uses the ZHD as a constant from any metrological source (blind models, VMF1 or any other model that could provide the ZHD). While the alternative approach does not use the metrological sources, instead it estimates a value for ZHD based on an initial value for the ZHD based on EKF.(We added explained this in section 2.3).

(2) The reference solution is a DD solution (Table 3). This DD solution follows the conventional approach, i.e. the a prior ZHD (from VMF1G) is fixed and the

ZWD is estimated. So, the reference for your alternative PPP approach is a conventional DD approach. Please write it down in section 2.2.

Authors' response:

It is true that the DD is the reference solution for validating the alternative and testing the conventional approach. In addition, the DD solution is based on VMF1G. However, the Global DD solution was used as a reference value because it could estimate the true values for the ZTD because of the long baselines used for adjusting the GPS observations. Meaning that if the DD solution based on short baselines, it cannot provide a reliable ZTD and because of the similarity of the ZTD for the nearest points. ("We used the Global DD solution as a reference value for comparing the ZTD because it could provide a reliable reference value based on long baseline as well as the initial values from VMFG1." Added to section 2.2)

(3) Table 5 (the conventional approach) can be directly compared with table 7 (the alternative approach). Where do the huge biases and rmse (6 cm) in table 5 come from? A blind model, e.g. the GPT2w, is much better (Table 1).

Authors response:

Using the conventional approach the ZHD will be used as constant value from the used mode (e.g. GPT) and because we estimate the ZWD it cannot overcome the difference and absorb the bias in the ZHD from the model. The rms in the Table 5 is coming from the bias from the considered ZHD from the model.

(4) In the alternative approach you also make use of an a prior ZHD. You also mention that the constraints for the ZHD and ZWD must be chosen 'very carefully'. I would like to see the following experiment: repeat the processing with various constraints for the ZHD and add the results in table 7. In particular, if you use some reasonable a prior ZHD (from GPT) and apply a very tight constraint on the ZHD you should obtain the results in table 5. In addition, i suggest to add the results for various constraints in Table 9 and 10.

Authors response:

The chosen values for the constraints of the ZHD and ZWD is presented in this manuscript for evaluating only the ZTD and this what the manuscript and the validation about, as we mentioned in the last sentence of section 2.3 "Bear in mind that the validation will only be addressed by the ZTD components because at this stage we are not attempting to separately validate the two components." A future step in this research will to present some values that could be computed from real ZWD and ZHD from e.g. VMF model and a validating for the separated values ZWD and ZHD using a new datasets following the same strategy.

Minor comments:

Page 3, line 13: In this context (slant tropospheric delays) you should add and comment on the following recent AMT paper:

Kaˇcmaˇrík, M., Douša, J., Dick, G., Zus, F., Brenot, H., Möller, G., Pottiaux, E.,Kapłon, J., Hordyniec, P., Václavovic, P., and Morel, L.: Inter-technique validation of tropospheric slant total delays, Atmos. Meas. Tech., 10, 2183-2208,https://doi.org/10.5194/amt-10-2183-2017, 2017.

Authors' response:

"Kačmařík et al. (2017) presented results of validating tropospheric slant total delays obtained from GNSS data processing with those obtained from NWM ray tracing, WVR measurements and collocated GNSS stations, in search of the optimal method for estimating GNSS STDs and found that the majority of evaluated GNSS solutions used deterministic models with rather long validity of estimated tropospheric parameters for which the residuals are important to overcome modelling deficiencies of low-resolution parameter estimates in time." Added to the manuscript.

Page 3, line 21: '...All the tropospheric models used for providing the hydrostatic or wet components rely on measured data to predict the tropospheric delay. However, they cannot account for weather variation and thus, cannot provide highly accurate estimates of the tropospheric delay. Furthermore, none of the tropospheric models account for the

diurnal variations of the troposphere. For example, they assume that pressure will be stable for a particular day of the year and that...' Here you mean the blind tropospheric models mentioned in table 1? Please clarify this. Tropospheric models derived from weather models (VMF1, UNB-VMF1 etc.) or measurements take into account the diurnal variation, etc.

Authors' response:

"while tropospheric models derived from weather models (VMF1, UNB-VMF1 etc.) or measurements take into account the diurnal variation " Added to the manuscript.

Page 5, Table 2:

Troposphere mapping function: Simply write 'New Mapping Function (Niell, 1996)'. There is no need to explicitly state that it provides separate wet and dry MF. This is standard.

Authors' response:

Done.

Page 6, Table 3:

Troposphere: I suggest to replace 'a-prior modeling of troposphere effects using VMF1G and estimation using zenith path delay and gradient parameters.' by 'a-prior modeling of troposphere effects using VMF1G and estimating the zenith wet delay and gradient parameters.' This is the conventional approach. I also suggest to use the same items in Table 2 and 3 (one item for the tropsopheric mapping function and another item for the a-prior ZHD and ZWD).

Authors' response:

Done.

Page 7, equation 5 and 6: Replace 'dtrop' by 'T' (see equation 1 and 2) and replace 'dh' by ZHD and 'dw' by ZWD.

Authors' response:

Done.

Page 10, line 2: 'Table 5 shows the difference in the tropospheric models used for comparison with the ZTD from DD GPS. These results are consistent with the literature, e.g. Li et al. (2012) quote RMS differences of 5.4 cm, 5 cm and 4 cm for SBAS, UNB3m and IGGtrop respectively and Pace et al. (2010) quote residuals in the order 50-100 mm.' The results are inconsistent with many other studies. For example, the recent study by Dousa et al 2016 AMT, shows that ZTDs from DD and PPP agree very well. I suggest to comment on that and add the following recent AMT reference:

Douša, J., Dick, G., Kaˇcmaˇrík, M., Brožková, R., Zus, F., Brenot, H., Stoycheva, A.,Möller, G., and Kaplon, J.: Benchmark campaign and case study episode in central Europe for development and assessment of advanced GNSS tropospheric models and products, Atmos. Meas. Tech., 9, 2989-3008, https://doi.org/10.5194/amt-9-2989-2016, 2016.

Authors' response:

"However, the recent study by Dousa et al. (2016) who suggested that there is a potential for advanced GNSS tropospheric products for meteorological applications and emphasized a synergy in GNSS and meteorological data and products Dousa et al 2016 AMT, shows that ZTDs from DD and PPP agree very well and that because they used VMF1 as a priori value with 6 values during the day." Added to the manuscript.

Page 12, line 3: 'Similar PPP configurations were adopted, except that the tropospheric delay was estimated using the new strategy with initial values of 2.1 m and 0.1 m for the dry and wet components, respectively' Here, I strongly recommend to add additional experiments (see my major comment). For example, instead of 2.1 m and 0.1 m as initial values use the GPT, and instead of a very loose constraint for the ZHD use a very tight

constraint for the ZHD, i.e. fix the ZHD. This procedure shows the transition from the alternative to the conventional approach.

Authors' response:

Currently we are working on evaluating this approach using an initial value from different sources. For example, VMF1 and many other models. Also, we are working on providing a reliable process noise based on 10 years data from VMF1 based on the Latitude and Longitude of the observer. In this manuscript we are presenting the possibility of this alternative approach to provide the ZTD without using a model and how could this approach be used in real time with the suggested process noise for obtaining the correct ZTD and we mentioned that in the manuscript that we will not evaluate nor validate the separated values of the ZTD as we stated in section 2.3 "Bear in mind that the validation will only be addressed by the ZTD components because at this stage we are not attempting to separately validate the two components.". In our new project the correct ZHD and correct ZWD will need the correct initial values from the models.

Anonymous Referee #3

General Comments

This paper intents to describe an alternative strategy to the conventional one used to estimate ZTD. The proposed alternative strategy is interesting and deserves to be evaluated. However, I think that the manuscript needs major revision before being ready for publication. The following aspects need to be improved. First, throughout the manuscript there is a lack of explanation of the obtained results. They are mainly presented in form of tables and figures with very short and poor text of critical discussion. Second, the proposed alternative strategy is supposed to improve both post–processed and Real-Time ZTD estimation. Post-Processed and Real-Time analysis have different requirements in term of latency and accuracy that should be discussed and considered. I have the feeling that the in the manuscript post-processed, near-real time

and real time issues, along with the related products used for data reduction and evaluation, are sometimes mixed up.

**Below specific comments for each section of the manuscript.**

Abstract Page 1 - line 17: Delete 'GPS' before ZTD Page 1 - line 18 Delete 'PPP' before GPS, GLO and GPS+GLO. How the differences are computed? Is it 'DDPPP' or 'PPP-DD'?

Authors' response:
Done, and the differences were computed in terms of DD-PPP.

Page 1 - - line 19- 21: 'Validation was also performed through comparison with the IGS ZTD values, for 12 weeks, with an overall RMS of 5.9 mm and against IGS real-time products with an overall RMS of 8.1 mm' I think that in the first part of this sentence the authors are referring to 'IGS final ZTD estimates' (http://www.igs.org/products) while in the second part to the 'IGS real-time orbit and clock products' (http://www.igs.org/rts/products) delivered in the framework of the IGS Real-Time Service and used for Real-Time PPP. If it is so, the sentence has to be properly rewritten. Anyway a clarification is necessary.#

Authors' response:

Done with a clarification for the sentence.

Introduction Page 2 - line 15. '. . ..an agreement of 1.7 cm' in terms of mean or standard deviation?

Authors' response:

It is RMS and a clarification has been made for the sentence.

Page 2 – line 27. Use the reference: B. Pace, R. Pacione, C. Sciarretta, G. Bianco, "Computation of Zenith Total Delay Correction Fields using Ground-Based GNSS estimates", IAG Symposia Series. vol 137-2012/ IAGS-D-13-00021 instead of Pace et al. (2010).

Authors' response:

Done, the reference has been replaced and added to the manuscript.

Page 3 – Line 1-7. The authors should add a reference of the cited models along with a brief description of their characteristics and differences in order to let the reader to understand the results presented.

Authors' response:

"Collins and Langley (1996) developed the UNB3 model for Wide Area Augmentation  System users. In the UNB3 algorithm, a look-up table of five atmospheric parameters (pressure, temperature, water vapor pressure, temperature lapse rate, and water vapor pressure height factor) that vary with latitude and day of year is used to calculate the surface meteorology.

UNB3m is a modified version of University of New Brunswick's (UNB3) neutral atmosphere model. It was created by altering a parameter's values in the UNB3 look-up table and the associated UNB3 algorithms (Leandro et al., 2006)." Added to the manuscript.

Page 3 - Line 19. I suggest replacing Table 1 with a brief summary of the results of Böhm et al. (2014) that are relevant to the present manuscript.

Authors' response:

"they have introduced a new blind tropospheric delay model which is based on gridded values of water vapor pressure, water vapor decrease factor, and weighted mean temperature. In terms of zenith total delays, the globally averaged bias is below 1 mm

and the RMS difference is about 3.6 cm as when compared to zenith total delays from GNSS at 341 globally distributed sites. Since GPT2w is also equipped with fully consistent hydrostatic and wet VMF1 coefficients, it may not only be used for positioning and navigation purposes but also for high precision applications, like geophysical studies, where the wet mapping functions are essential to estimate residual zenith wet delays. GPT2w also contains the mean values as well as annual and semiannual amplitudes of the weighted mean temperature. This is an important quantity for the determination of the integrated water vapor or precipitable water as required in GNSS meteorology (Bevis et al. 1992)." Added to the manuscript and we have deleted Table 1.

Page 4 - Line 8. '..static and real-time situation' I think it is post-processed and real-time situation.

Authors' response:

Done.

Page 4 - Line 10. Sentences on the organization and structure of the paper must be added.

Authors' response:

"Next sections presented the methodology of PPP solution and DD solution as a reference." Added to the text.

PPP Daily Solution Methodology Page 5 - Line 15. Table 2. Please consider that:

1. ANTEX from IGS the proper reference to the ANTEX file is M. Rothacher, R.Schmid: ANTEX: The Antenna Exchange Format, Version 1.4, 15 September 2010, ftp://igs.org/pub/station/general/antex14.txt Which antex file is used? It the same antexfile used to process DD data of Table 3 in section 2.2?

Authors' response:

ANTEX file is antex08 that has been used for the processing in both DD and PPP.

2. Troposphere: Using Saastamoinen model for the hydrostatic component and estimate the wet as a state, unless otherwise mentioned. Please explain the meaning of 'state'. What is the ZTD sampling rate?

Authors' response:

State mean unknown (or state in the extended Kalman filter). The sampling rate of the ZTD means the interval of the estimated values of the ZTD and it is the same sampling rate of the rinex data.

3. Troposphere mapping function: drop 'new'

Page 6 - Line 9. Table 3. Please consider that: 1. Troposphere. What is the ZTD sampling rate?

Authors' response

The sampling rate of the ZTD means the interval of the estimated values and it is with an hourly sampling rate.

There are several differences in the models summarized in table 2 and table 3: mapping function, carrier phase ambiguities, and products. Please add a comment on these diversities and on their potential impact on the estimated ZTD.

Authors' response

"There are many differences between the PPP strategy and network double difference strategy. There is a potential that these diversities could affect some of the estimated parameters e.g position component. However, a 24 hour of data processed using any of these strategies could minimize their potential impact of the estimated ZTD. Also, an estimated value of an hourly ZTD could also minimize that diversities. The remaining parameter that could affect the estimated ZTD could be the ambiguities and this has been handled following a float solution in the PPP processing so that any differences will be absorbed by the estimated float ambiguities." Added to the manuscript.

New Strategy for Estimating Tropospheric Delay Page 7 - Line 18-19. On which ground the values for the random-walk process noise were chosen? Did the authors test other values?

Authors' response

The values of the random-walk process noise were chosen arbitrary for one station, with known values of ZTD from Global Double Difference solution. Then start to change those values of the process noise till we got the best agreement for both ZTDs from the PPP solution and the DD solution. Finally, we generalized and fixed those values for all the dataset in this manuscript. Thus, a good process noise for the separated values of the ZTD naming ZHD and ZWD need to be tested similarly with the available reference values for ZHD and ZWD. And that is the reason why we did not test or validate any of the separated values of the ZTD ( ZHD nor ZWD).

Datasets for Comparing the Two Methods Page 8 - Line 10.'. . . with optimum 24-h observations' What do you mean with optimum?

Authors' response

That datasets were downloaded as an hourly rinex files as they were uploaded by the OSGB to the server. Some stations have problem of maintaining or delivering all of the 24 rinex files to the server. Thus, excluded any station that did not have a 24 rinex file for the day (optimum 24-h).

Results using the Traditional Strategy Page 9 - Line 9. Is it 'static' or 'post-processed'?

Authors' response

The results using Traditional strategy is a post-processed and static solution.

Page 9 - Line 17. I suggest to consider also the site coordinate repeatability as an internal quality metric to check the different solutions.

Authors' response

We have computed the coordinate repeatability for all PPP scenarios using the four tropospheric mode and included that in Table 4 (previously Table 5)

| Coordinate repeatability | | |
|---|---|---|
| E | N | U |
| GPS | | |
| 2.7 | 2.4 | 5.5 |
| 2.4 | 2.6 | 5.3 |
| 2.6 | 2.5 | 5.5 |
| 2.4 | 2.6 | 5.4 |
| GLO | | |
| 2.1 | 4.3 | 9.1 |
| 2.0 | 4.4 | 9.2 |
| 2.1 | 4.4 | 8.6 |
| 2.1 | 4.4 | 8.9 |
| GPS+GLO | | |
| 2.0 | 3.2 | 5.9 |
| 2.1 | 3.3 | 6.1 |
| 2.0 | 3.3 | 5.5 |
| 2.1 | 3.2 | 5.5 |

Validation for One Continuous Week Page 10 - Line 19. Typo 'troposphelric'

Authors' response

Done.

Page 10 - Line 24. The authors are testing the new strategy over 1 week continuous solution, they are not averaging 7-daily solutions to get the weekly solution. A comment on how the GNSS orbits are handled is required since GNSS orbits over 1 week are available as 7 independent daily solutions with a possible jump at midnight between consecutive days.

Authors' response

We agreed with the reviewer that one continuous week solution is not the averaging of 7-daily solutions. "The precise orbit files which contain the satellites coordinate every 15 minutes were interpolated in our software to provide the satellites coordinate based on the data rate (rinex rate). This could minimize the possible jump at midnight between consecutive days." Added to the manuscript.

Page 11 - Line 1. Table 6. The ZTD agreement is -6.3 mm mean and 5.8 mm std. Is this a 'good' agreement' or there is room for improvement? In post-processing the expected agreement, in terms of std, between different sw/solutions is about 2 mm. (see Pacione et al: Atmos. Meas. Tech., 10, 1689-1705, https://doi.org/10.5194/amt-10-1689-2017, 2017)

Authors' response

a standard deviation of 5.8 mm could be treated as a good agreement between a PPP solution without using any tropospheric model nor information about the weather and the Global Double Difference solution as well as -6.3 mm because those differences will be neglectable when will be converted to the IWV as suggested by Dousa and Vaclavovic (2014) that there result were within 6-10 mm ZTD mean standard deviations which can fulfil the requirement for the operational NWP (i.e. 30mm ZTD).

In the mentioned paper by all the processing were following the network double difference solution except GIPSY solution mentioned in the paper as AS0 which was using PPP solution. Conserving the PPP solution from GIPSY they were using their own JPL precise products and that may give the consistency mentioned by the reviewer. While in our PPP solution we are using our software that is not capable at this stage of producing the precis products.

Page 11 - Line 7. Fig. 2. Could the authors explain why PPP ZTD is systematically larger than DD ZTD? Are there any boundary problems in the PPP ZTD at the beginning and end of the 7-day period?

Authors' response

There were no boundary problems in the PPP ZTD. However, the ZTD estimated from our PPP solution is provided in 30-second interval while it is 1-hour interval from the Global Double Difference.

Long-Term Validation of the New Strategy Page 13 - Line 9 and Fig. 3. To demonstrate that there is no seasonal behavior in the proposed strategy, one average value for the whole year is not enough. One value per month or one value for season should be considered.

Authors' response

We provided a one value per season for all the statistics and all PPP scenarios. Figure 4 is included now to summarise that and text below has been modified.

"To illustrate that the new strategy may be used at any time of the year, irrespective of the weather conditions, Figure 4 considered one value (Mean, RMS and STD) for season using the new strategy for the same dataset in Figure 3."

Global Validation using IGS Stations Page 14 - Line 11-13. Could you please better explain which products you used? According to what listed in http://www.igs.org/products IGS is not delivering NRT ZTD.

Authors' response

It is available here ftp://cddis.gsfc.nasa.gov/pub/gps/products/trop_zpd

Global Validation using Real-Time Products Page 16 - Line 5. I suggest the following references for Real time ZTD estimation and performance 1. Dousa J, Vaclavovic P (2014) Real-time zenith tropospheric delays in support of numerical weather prediction applications. Advances in Space Research (2014), Vol 53, No 9, pp 1347-1358, doi:10.1016/j.asr.2014.02.021 2. Ahmed F et al Comparative analysis of real-time precise point positioning zenith total delay estimates, 2014 GPS

Solut. DOI 10.1007/s10291-014-0427-z

Authors' response

"In some aspects, these results are consistent with Dousa and Vaclavvovic (2014), who obtained 6 to 10mm standard deviation in tropospheric ZTD, but with large biases of up to 20mm, which they suggested were due to missing precise models in their software. While these results are significantly better than Ahmed et al. (2014) who obtained a 1 to 3cm bias and 1 to 4 cm standard deviation in tropospheric ZTD." Added to the manuscript.

Conclusions Page 20 – Line 1-4. I think that what the authors are assessing is very strong. The new proposed strategy is interesting but needs to be checked and tested more deeply before drawing this conclusion.
References When possible, replace the reference listed as 'proceeding' with a peer review publication.

Authors' response

We have updated the references list based on your recommended references as well as the recommendation from the other two referees.

Other comments: 1. All the acronyms in the text have to be explained and the same abbreviation should be used to refer to the same thing, for example sometimes is used 'GLONASS' sometimes 'GLO'. 2. In the tables the sign of the considered difference has to be reported.

Anonymous Referee #2

Major remarks and questions:
1. Authors presented results of ZTD estimation obtained using alternative strategy of PPP method and compered these results to ZTD obtained using double

differed method. Why Authors did not compere their results also to the conventional PPP?

Authors' response

The results presented using the conventional approach (using a model for ZHD and estimate the ZWD as an unknown) is presented in section 3.1 and all the results summarized in Table 5 comparing to the Double difference solution as a truth values. Also, the results using the alternative approach has been compared to the same reference solution (the double difference) this means that we want to understand which one from the two methods has a better solution for both the position solution as well as tropospheric products. If we compared the two solutions (the alternative approach and the conventional method with each other, we will end up with a comparison that needs to be evaluated. This is because our solution from the conventional approach is not the reference solution to be compared with. Meaning that we need a reliable solution to compare our two solutions.

2. Line 24-26: It is not entirely true. Authors should remember about VMF, where different values of ZHD and ZWD are provided during the day.

Authors' response

'except VMF1 which provides 4 values during the day and real-time PPP services may not want to use VMF1 as this would require more information to be passed via the communications satellite.' Added to the text.

3. In my opinion, in the introduction there is lack of information about VMF. Authors can also pay attention to VMF3/GPT3 (doi: 10.1007/s00190-017-1066-2).

Authors' response

"Landskron and Böhm (2017) presented a refinement to the Vienna Mapping Function 1 (VMF1) which is referered to as Vienna Mapping Function 3 (VMF3). To eliminate the shortcoming in the empirical coefficients b and c. They also presented a new empirical mode Global Pressure and Temperature 3 (GPT3)." Added to the introduction section.

3. Table 2: Which ANTEX version was used by the Authors? Why Authors did not fixed ambiguities?

The Antex version was ANTEX08 "Applied using the ANTEX08 file from IGS depending on the GPS week, to be compatible with the products." Added to the text

4. In PPP solution Authors used precise products from NRCan while in DD products from CODE. Products from different sources can caused differences in parameters estimation, which may lead to wrong conclusions. Ephemerides and clocks should be used from the same sources, either from CODE or from NRCan. In my opinion it will be nice to see comparison between two PPP methods: first with conventional strategy, and second with alternative strategy proposed by the Authors. Of course in both solutions the same processing parameters and precise products should be used.

Authors' response

The precise products take a big control for the PPP solution. We have chosen this product because it provides the GLONASS products as well for that time of the data and the processing. CODE products were not included the GLONASS precise products at the time of the processing. In addition, those product (EMX) was proofed to be very stable and give a reliable solution as presented in Jareer et.al for comparing the PPP GLONASS solution which is a major solution in this manuscript.

Moreover, Authors should remember that for precise tropospheric parameters estimation the VMF1 is often used. Unfortunately Authors did not provide any information about it and did not used solution with VMF1 to comparison (e.g. in Table 5). I think that this is a big deficiency of presented manuscript. In the paper it is hard to find explanation for this, especially that VMF1 was used in DD solution.

Authors' response

In PPP solution the ZTD is produced based on the data rate (30 sec in our solution and data) while VMF1 is provided every 6 hours which means four values during the day. So it is not logical to compare our PPP ZTD with that. A more reliable solution to be compared with is the DD solution that is based on the VMF1 as an initial solution to

provide a more resolution solution (1 value every hour) which will give us a 24 values for the comparison.

The second reason why we did not used the VMF1 for comparing our solution is the VMF1 is available in a grid form, and this needed to be interpolated for the selected stations in the UK which will reduce the accuracy of the comparison. While the DD solution provided the ZTD for the stations using their rinex file rather than an interpolated one.

5.  One more question related to solutions using traditional strategy. In section 3 Authors presented used tropospheric models. Why Authors did not used GPT2 which is more precise than other presented models? Of course the VMF case should be also reconsider in this place.

Authors' response

The four models presented in the manuscript are the models that has been implemented in our software (POINT software) and this is the reason why we have used only those four models. And the reason for not using VMF1 is that we have focused only on the hard-coded models without relying on any other input from the online services so that it can be applicable in the real time solution.

7. I have also general comment to the all results presented in tables: 5, 6, 7, 9, and 10. I think that in case of ZTD differences it will be nice to see also maximum and minimum values. I also advise Authors to think about replacing mean values into a median.

Authors' response

We could use the median in case of our solutions have a non-normal distributed value as provided in the figure below.

[Figure]

8. In validation section it is not clear how the ZTDs from IGS and Authors strategy were compered. In IGS tropospheric products are available with 5 minutes interval. I think that information about estimation interval should be placed into manuscript (both for IGS and Authors solution), as well as information how the comparison of two products looked like. Was ZTDs comparison conducted at the same epochs or maybe some averaging was used? It should be clearly explained in the text.

Authors' response
The 5 minutes ZTD solution from IGS is not the final solution from IGS> the comparison of the products was performed using the common epoch without any averaging method. This has been mentioned in the manuscript page 9 line 14-15, page 14 line 15.

9. Section 6.1: I have serious objections to the content of this section. Authors show how the 'new strategy' impacts on convergence time. However they only presented

comparison to the low precise tropospheric models. I think that it is necessary to present results to more precise solution e.g. with VMF.

Authors' response

We presented our solution as the four models comparing to the new strategy, meaning that if we use the new strategy what is the improvement comparing to what have been used now using the conventional approach.

Furthermore, Authors did not provide any information about processing strategies, or number of used stations. Readers may also have a problems with results interpretation. Are there mean value of convergence time in Figure 7? Or maybe these values are for one station? If there are mean values, Authors should present also RMS or STD parameters.

Authors' response

In section 6.1 page 18 line 13 we mentioned that the dataset for this convergence time analysis are in Table 4 (now it is Table 3) and the processing strategies was the same that had been used in Table 5 (now it is Table 4). Meaning that this convergence time analysis is based on (523(summation of the used stations per week) * 7 days) = 3661 procced site.

The figure represents the improvement in the PPP coordinates convergence time when using the new strategy comparing to the four models. Meaning that it is the difference between the mean of all convergence time for each PPP solution applying a tropospheric model (for all stations mentioned in Table 3) and the same mean when applying PPP solution using our tropospheric strategy for the same data sets.

10. Section 6.2: In this section Authors compared only for two stations. Why exactly these? Authors present results for conventional and alternative approaches. Which tropospheric model was used in the conventional solution?

Authors' response

Using Saastomoniane tropospheric model. We have chosen those stations because they have the most continuous data comparing to the other station in the UK.

11. In presented by Authors solution the ZHD and ZWD components are estimating separately. In presented manuscript Authors presented only total value of ZTD. However this not mean that ZHD and ZWD are correctly estimated. For example there can be some biases for both component but with opposite signs. Thus in ZTD this error will not be visible. It should be notice that proper estimation of ZWD is crucial for many application, e.g. for conversion to IWV and analysis of atmospheric opacity is performed (doi: 10.1007/s10291-017-0675-9). Unfortunately in presented manuscript there it is not explained whether the ZWD can be directly used for such (or similar) application.

Authors' response

In the alternative strategy, and at this stage of the research we do not provide any information about the ZHD nor the ZWD because as the referee said they could balance that error in the opposite sign. we are working on extend this strategy to separate the ZWD and ZHD from the ZTD using a different external models and numerical weather models.

12. In presented manuscript there is also lack of ZHD and ZWD estimation errors analysis. I think that it is necessary to show how the values from covariance matrix looks like during the processing time. Also it will be nice to see post-fit residuals. Of course, only examples for selected stations can be presented. The proposed by Authors strategy is very interesting but it needs to be checked, tested, and verified more deeply before publication.

Authors' response

As we said in the previous reply we cannot provide or trust the separated values at this stage nor on this data set> What we are presented in this manuscript is the capability of this strategy to provide an accurate ZTD with an accurate position component comparing the Network DD solution. This research could and will be extended to evaluating the ZWD and ZHD based on the numerical weather modelling and VMF models as well as an extended application in the for the real time PPP solution.

COLLINS, P. & LANGLEY, R. 1996. Limiting Factors in Tropospheric Propagation Delay Error Modelling for GPS Airborne Navigation‡. *The Institute of Navigation 52nd Annual Meeting.* Cambridge, Massachusetts, USA.

KAČMAŘÍK, M., DOUŠA, J., DICK, G., ZUS, F., BRENOT, H., MÖLLER, G., POTTIAUX, E., KAPŁON, J., HORDYNIEC, P., VÁCLAVOVIC, P. & MOREL, L. 2017. Inter-technique validation of tropospheric slant total delays. *Atmospheric Measurement Techniques,* 10**,** 2183-2208.

LEANDRO, R., SANTOS, M. & LANGLEY, R. UNB Neutral Atmosphere Models: Development and Performance.  Proceedings of the 2006 National Technical Meeting of The Institute of Navigation, 18-20 January 2006 Monterey, CA. 564-73.